# Possibility for strong northern hemisphere high-latitude cooling under negative emissions

Jörg Schwinger [1✉], Ali Asaadi [1], Nadine Goris [1] & Hanna Lee[1,2]

It is well established that a collapse or strong reduction of the Atlantic meridional overturning circulation (AMOC) would substantially cool the northern high latitudes. Here we show that there is a possibility that such cooling could be amplified under deliberate $CO_2$ removal and result in a temporary undershoot of a targeted temperature level. We find this behaviour in Earth system models that show a strong AMOC decline in response to anthropogenic forcing. Idealized simulations of $CO_2$ removal with one of these models indicate that the timing of negative emissions relative to AMOC decline and recovery is key in setting the strength of the temporary cooling. We show that the pronounced temperature-fluctuations at high northern latitudes found in these simulations would entail considerable consequences for sea-ice and permafrost extent as well as for high latitude ecosystems.

[1] NORCE Climate, Bjerknes Centre for Climate Research, Bergen, Norway. [2] Department of Biology, Norwegian University of Science and Technology, Trondheim, Norway. ✉email: jorg.schwinger@norceresearch.no

Since the Paris agreement entered into force in 2016, global anthropogenic carbon dioxide emissions have increased by 5% (0.58 Pg C between 2016 and 2019)[1], in stark contrast to the fact that net zero $CO_2$ emissions are required by around 2050 to meet a 50% chance of keeping the average surface temperature increase to below 1.5 °C[2]. A temporary exceedance or overshoot of the Paris Agreement temperature targets and subsequent carbon dioxide removal (CDR) from the atmosphere[3,4] is therefore increasingly discussed as an option to delay the costly transition to a zero-carbon economy and society. Indeed, climate model experiments with application of CDR[5,6] support the feasibility of such a strategy, since they show that many aspects of climate change are largely reversible if certain thresholds are not crossed. On the other hand, overshoot strategies come with adverse effects related to slowly reacting components of the Earth system, for example, permafrost[7,8] or the interior ocean[9,10]. Since it is politically attractive to delay mitigation action, it has been argued that limits of overshoot scenarios beyond which adverse effects outweigh advantages need to be investigated and defined[11].

In this work, we have simulated temperature overshoots of different magnitude and duration using a state-of-the-art Earth system model (NorESM2-LM[12,13], methods) based on idealized, bell-shaped emission curves that have been developed for the Zero-Emission Commitment Model Intercomparison Project (ZECMIP)[14]. As a reference case for our study, we take a simulation that is roughly consistent with the goals of the Paris agreement leading to a temperature increase of about 1.7 °C in the long term with no CDR applied. This reference simulation (referred to as $B^{1500}$) has 1500 Pg of cumulative carbon emissions during the first 100 years and zero-emissions thereafter (Fig. 1a). Additionally, we perform three simulations with the same emission profile but higher carbon emissions ($B^{1750}$, $B^{2000}$, and $B^{2500}$, the superscript indicating the amount of cumulative carbon emissions in Pg). These simulations reach peak temperatures between 2.2 and 3.0 °C during the positive emission phases (Fig. 1c and Table 1). From these simulations, we branch off negative emission phases of 250, 500, and 1000 Pg cumulative carbon removal, applied over a time span of 100 years. At the end of the negative emission phases, all simulations have experienced cumulative carbon emissions of 1500 Pg C (Fig. 1a). To simulate overshoots of different duration, we vary the branch point in time such that there are either 0 or 100 years of zero-emissions before CDR is applied. Thus, we get six different overshoots relative to the reference simulation, which we refer to as $OS_0^{250}$, $OS_{100}^{250}$, $OS_0^{500}$, $OS_{100}^{500}$, $OS_0^{1000}$, and $OS_{100}^{1000}$ (Table 2). Here, the superscript refers to the cumulative amount of CDR applied, and the subscript refers to the length of the zero-emission phase before the start of the simulated CDR. We stress that our simulations are idealized. For example, the phase of positive emissions is limited to 100 years, which is substantially shorter than the period from 1850 to 2100 usually covered in historical and scenario simulations with Earth system models (ESMs). In addition, we do not consider forcing from anthropogenic aerosols and greenhouse gases other than $CO_2$. We note, however, that our simulations are closer to a realistic scenario than overshoot experiments based on a fixed rate of $CO_2$ increase and a mirrored decrease back to preindustrial levels, which have been used extensively[5,10,15–17]. The latter setup results in an extremely abrupt change from positive to negative emissions, and the amount of implied negative emissions is generally huge. Our simulations have a period of increasing emissions followed by a period of decreasing emissions, and the positive and negative emission phases are smoothly joined. Further, $CO_2$ concentrations are not returned to preindustrial levels, but to a level compatible with keeping global temperature increase well below 2 degrees. We investigate the robustness of results from our idealized model experiments by comparing with results from other types of simulations (including a more realistic scenario simulation) conducted with our model and other state-of-the-art ESMs.

## Results and discussion

**Positive and zero-emissions.** The strength of the Atlantic meridional overturning circulation (AMOC) declines by 48 to 76% of its preindustrial strength (Fig. 1b and Table 1) during the positive

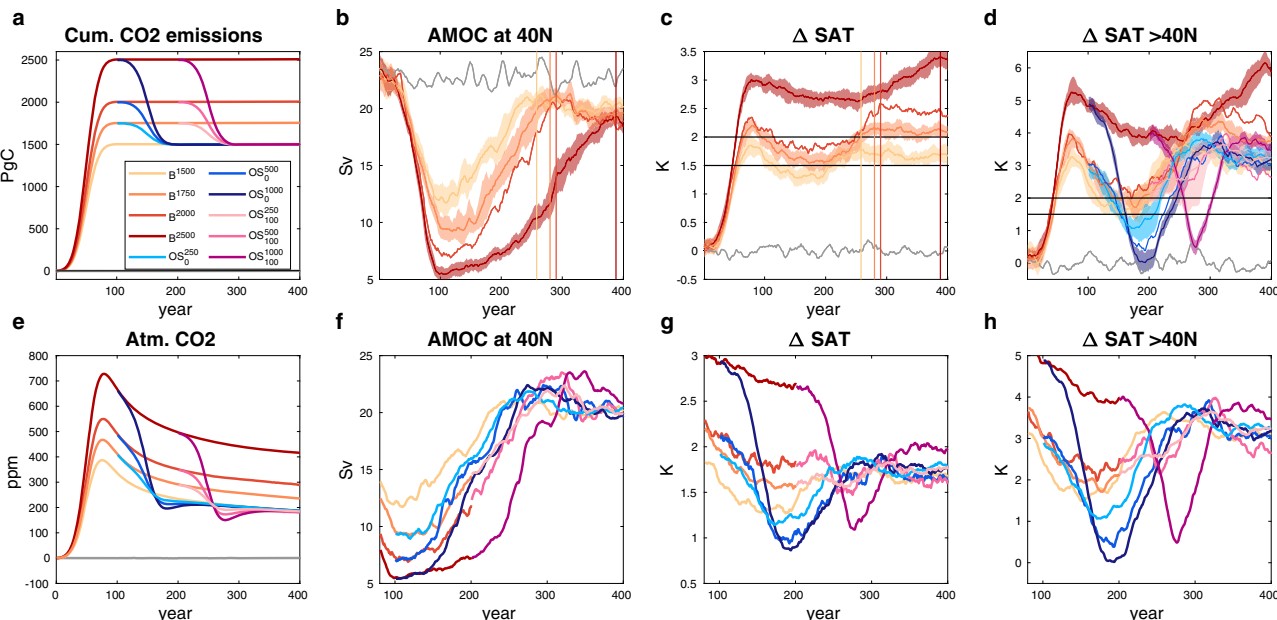

**Fig. 1 AMOC and surface temperature changes in idealized NorESM2 simulations. a** Cumulative carbon emissions, **b**, **f** AMOC strength at 40°N, **c**, **g** global average surface air temperature (SAT) change relative to preindustrial conditions, **d**, **h** change in average SAT north of 40°N, and **e** atmospheric $CO_2$ concentrations for the B- and OS-simulations. Color coding of simulations is indicated in the legend in panel **a**. If available, the mean and range (shaded areas) of three ensemble members are shown. For clarity, panels **b** and **c** only show the B-simulations that have positive and zero-emissions. Likewise, panels **f**–**h** only show the ensemble means of the overshoot simulations (blue and magenta lines) branched from the B-simulations (yellow to red lines). In panels, **b**–**d** and **f**–**h** an 11-year running mean has been applied to the data.

**Table 1 Overview of B-simulations: B-simulations have positive emissions during the first 100 years and 300 years of zero-emissions thereafter.**

| Simulation | Cumulative emissions | Ensemble members | Peak SAT[a] | SAT decline after peak[a] | Final SAT[a] | AMOC decline[ab] |
|---|---|---|---|---|---|---|
| B[1500] | 1500 Pg C | 3 | 1.86 °C (1.85–1.95 °C) | 0.59 °C (0.60–0.75 °C) | 1.70 °C (1.60–1.78 °C) | 11.0 Sv (48.4%) (10.6–12.0 Sv; 46.7– 52.7%) |
| B[1750] | 1750 Pg C | 3 | 2.20 °C (2.15–2.26 °C) | 0.66 °C (0.57–0.81 °C) | 2.09 °C (2.07–2.13 °C) | 13.5 Sv (59.5%) (13.1–14.2 Sv; 57.6– 62.5%) |
| B[2000] | 2000 Pg C | 1 | 2.35 °C | 0.63 °C | 2.36 °C | 15.8 Sv (70%) |
| B[2500] | 2500 Pg C | 3 | 3.01 °C (2.99–3.08 °C) | 0.39 °C (0.40–0.55 °C) | 3.37 °C (3.31–3.43 °C) | 17.3 Sv (76%) (17.2–17.6 Sv; 75.7–77.5%) |

Emissions follow a bell-shaped curve peaking at year 50 as described in ref. [61].
[a]Where available, the ensemble mean of the 11-year running mean, the ranges given are minimum and maximum values for individual ensemble members.
[b]Preindustrial AMOC strength at 40°N is 22.7 Sv in our model.

emission phases. During the zero-emission phases of the B-simulations, AMOC strength gradually recovers and converges towards a level that is similar albeit lower by 3 to 5 Sv than the unperturbed state, consistent with results of a previous study[18]. We note that the simulated contemporary AMOC strength at 26°N in our model is 21 Sv and compares reasonably well to the observation-based estimate of 17.4 Sv[19]. NorESM2-LM is among the 4 CMIP6 models (disregarding one outlier model) with the highest contemporary overturning strength (CMIP6 range: 9.6 to 23 Sv; CMIP6 mean: 17.7 Sv)[19]. The model simulates the depth of the AMOC maximum at 0.9 km, slightly better than the CMIP6 model mean (0.84 km), but still shallower than the observation-based estimate (1.04 km)[19]. Our model generally shows a pronounced decline of AMOC strength with climate change. We discuss this point further below, where we compare the AMOC weakening in our model to other CMIP6 models.

Global average surface air temperature (SAT) and atmospheric $CO_2$ concentrations peak about 15 years before emissions cease (Fig. 1c, e). Thereafter, SAT declines for about a century by 0.39 to 0.66 °C (Table 1) before recovering and staying approximately constant until the end of our simulations (Fig. 1c; the steady-state is not reached for the B[2500] simulation). The cooling during the zero-emission phase is centered around the North Atlantic (Fig. 2a) and shows the characteristic footprint of a collapsed or strongly reduced AMOC[20–24]. The time of SAT recovery coincides well with the recovery of AMOC strength (colored vertical lines in Fig. 1b, c). Consistent with previous studies[21,22], the mechanism of the surface cooling north of 40°N is a strong reduction of latent heat fluxes to the atmosphere in the North Atlantic (Fig. 2c), resulting in a reduced heat loss from the ocean in this region. Atmospheric specific humidity decreases (Fig. 2e), which exerts a radiative cooling effect[21,22]. Since the cooling outweighs the reduced moisture flux from the ocean, low cloud cover increases (Fig. 2g), which leads to additional radiative feedback that cools the surface (Fig. S1).

**Negative emissions**. For this study, we take the long-term global SAT of the reference simulation B[1500] (1.7 °C above preindustrial temperature) as consistent with the Paris agreement. The goal of CDR is then to lower global SAT from the levels found in the B[1750], B[2000], and B[2500] simulations to the target level of the Paris agreement. However, if net negative emissions are applied immediately after positive emissions have been phased out (i.e., in our short overshoot simulations OS$_0^{250}$, OS$_0^{500}$, and OS$_0^{1000}$, blue lines in Fig. 1), northern high latitudes cool considerably below the target level before temperature rises again and approaches the target temperature from below (Fig. 1g, h). This effect is weak at latitudes south of 40°N (Fig. S2), but particularly strong north of 40°N. Here, the OS$_0^{1000}$ simulation even reaches preindustrial SAT levels for a short period of time before bouncing up to the reference level, which is ~3 °C above preindustrial temperature for this latitude band.

These results suggest that northern high-latitude cooling that is seen when emissions are phased out and that is caused by a strong reduction of AMOC can be amplified if CDR is applied subsequently. The mechanisms and feedbacks leading to this amplification are the same as explained above for the B-simulations (Fig. 2 and Fig. S1, left and right columns of panels). However, the negative emission phases of the overshoot simulations start from a state with a weaker AMOC compared to the reference simulation, since a larger amount of positive emissions leads to a stronger AMOC-reduction. The combination of a weaker AMOC state and a reduction of the radiative forcing by CDR amplifies the northern high-latitude cooling that is already present in the reference simulation.

**Table 2 Simulation overview for the negative emission simulations.**

| Simulation | Parent simulation | Cumulative positive emissions | Cumulative CDR | Time between positive and negative emissions | Ensemble members |
|---|---|---|---|---|---|
| $OS_0^{250}$ | $B^{1750}$ | 1750 Pg C | 250 Pg C | 0 years | 3 |
| $OS_{100}^{250}$ | $B^{1750}$ | 1750 Pg C | 250 Pg C | 100 years | 3 |
| $OS_0^{500}$ | $B^{2000}$ | 2000 Pg C | 500 Pg C | 0 years | 1 |
| $OS_{100}^{500}$ | $B^{2000}$ | 2000 Pg C | 500 Pg C | 100 years | 1 |
| $OS_0^{1000}$ | $B^{2500}$ | 2500 Pg C | 1000 Pg C | 0 years | 3 |
| $OS_{100}^{1000}$ | $B^{2500}$ | 2500 Pg C | 1000 Pg C | 100 years | 3 |

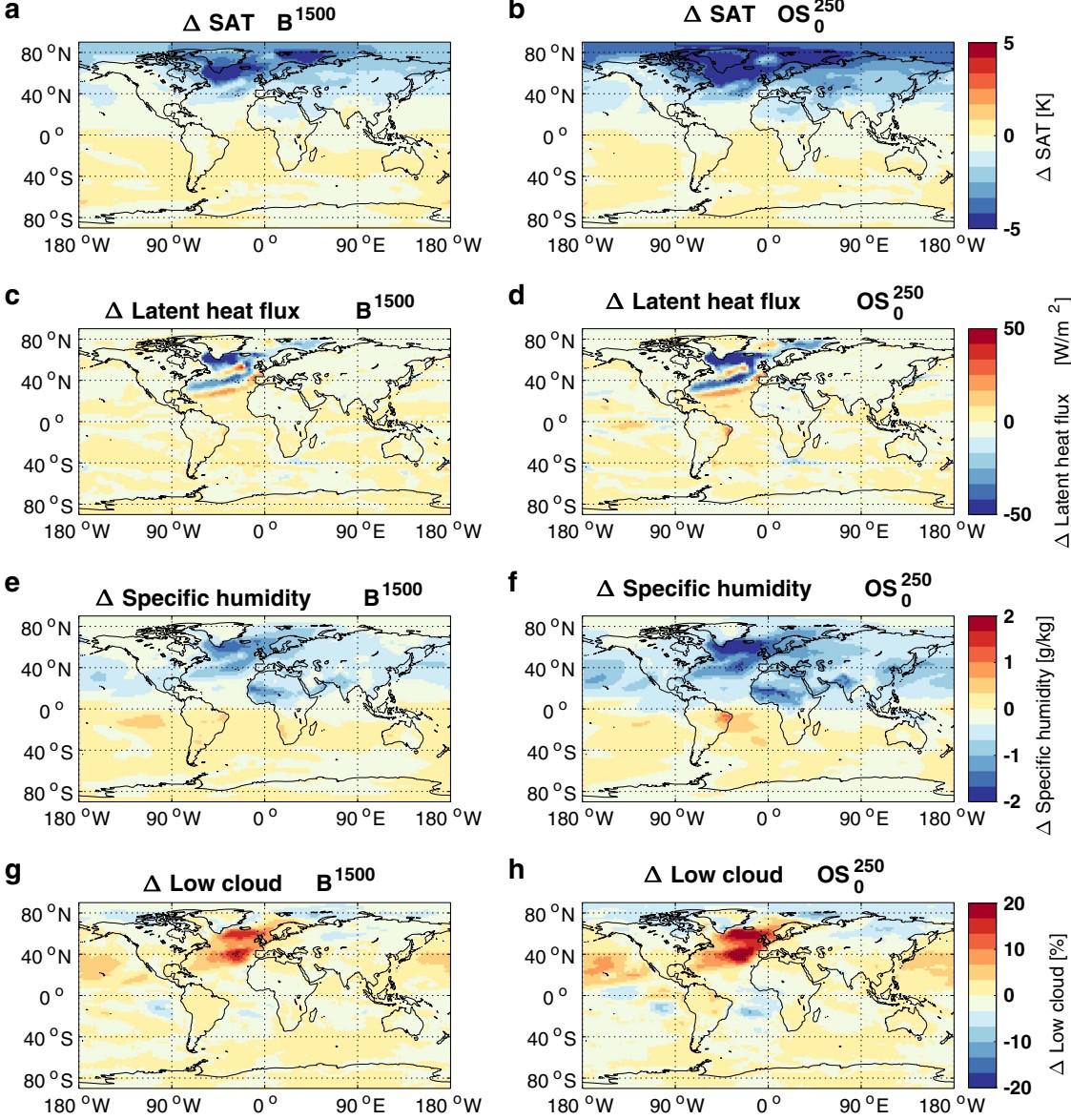

**Fig. 2 Impacts of AMOC decline under zero and negative emissions.** Difference between a state with reduced AMOC around the SAT minimum (years 180-190) and a state with recovered AMOC and SAT (years 280-289) for the reference simulation $B^{1500}$ (left column) and the $OS_0^{250}$ overshoot (right column). Shown is the mean of three ensemble members for **a**, **b** SAT, **c**, **d** latent heat flux, **e**, **f** specific humidity, and **g**, **h** low cloud cover.

Further analysis of the temperature response north of 40°N indicates that absolute temperature anomalies (relative to preindustrial temperature) in the overshoot simulations remain smaller over the ocean than over land, with a particularly strong cooling over the North Atlantic (Fig. S3a–c), as expected from the strong reduction of AMOC strength. Absolute surface

temperature anomalies in Europe are the smallest of all continents due to the proximity and downwind location relative to the North Atlantic. However, our analysis also shows that the additional cooling in the overshoot simulations relative to the reference (i.e., the simulation without overshoot) is regionally very similar north of 40°N (Fig. S3d–f). There is a tendency that

land masses located further downwind from the source of the anomaly in the North Atlantic experience less additional cooling during the overshoot, i.e. the additional cooling is weaker in North America than in Asia and weaker in Asia than in Europe. This effect can be clearly seen for the highest overshoot, but it is less pronounced in the low overshoot case (Fig. S3a, d). This indicates that the amplified cooling north of 40°N during an overshoot in our model is spatially very coherent with only minor regional differences.

If negative emissions are applied later, after 100 years of zero-emissions (Fig. 1, magenta lines), when AMOC strength is already recovering, the cooling effect of CDR is much weaker or even absent. In the $OS_{100}^{250}$ and $OS_{100}^{500}$ simulations, CDR starts at a point in time when AMOC strength has recovered to more than 50% of its original strength and SAT north of 40°N increases in the B-simulations (from which the overshoots are branched off) as well as in the reference simulation. In these cases, CDR mainly mitigates the increase of temperature rather than leading to cooling. Only the $OS_{100}^{1000}$-overshoot still shows a substantial cooling, consistent with the fact that its associated AMOC recovery is slower compared to the other simulations and SAT is still decreasing when CDR is applied.

**Impacts and challenges for adaptation**. If CDR is applied to lower surface air temperature, then why would we care if regional temperature undershoots the target level for some time, and could we not avoid such undershoot by regulating the pace of $CO_2$ removal? For the medium-sized overshoot $OS_0^{500}$, we find that the maximum cooling below the target level during the phase of negative emissions is −1.79, −0.31, and −0.12 °C, north of 40°N, between 40°S to 40°N, and south of 40°S, respectively (Fig. S2). These results illustrate that there could be tradeoffs or conflicts between the global north and south in a scenario of CDR deployment, specifically when avoiding a temperature undershoot below the target level in the north would require delaying CDR and accepting climate impacts in the south for a prolonged period of time. This is nicely illustrated by the long variants of the low and medium overshoots ($OS_{100}^{250}$ and $OS_{100}^{500}$) which provide a quite smooth temperature trajectory (with minimal undershoot) towards the reference simulation in the northern high latitudes and would certainly be considered an optimal CDR strategy for this region in isolation. Here, one would leverage the northern high-latitude SAT decline due to reduced AMOC strength until the recovery of AMOC increases SAT, which would then be offset by negative emissions. However, such a strategy would come at the cost of accepting elevated temperatures south of 40°N for an extended period of time.

An AMOC slowdown or collapse is thought to have severe impacts globally. Most notably, a southward shift of tropical rainfall and an increase of droughts in the Amazon and Sahel regions[25] would affect a large and vulnerable population. Here, we focus on the high-latitude temperature fluctuations seen in our overshoot simulations noting that the AMOC slowdown itself is in fact mitigated by the application of CDR in our model (i.e., AMOC recovery happens faster with than without CDR).

What would be the consequences of temperature fluctuations of the magnitude seen in the $OS_0^{250}$ and $OS_0^{500}$ simulations? The impacts of climate change on ecosystems and economies, as well as challenges for adaptation, are often discussed in terms of increasing global mean temperatures only. We anticipate that the northern high-latitude temperature trajectories in our overshoot simulations with their pronounced warming-cooling-warming pattern would most likely exacerbate impacts and challenges[26]. To illustrate this point, we pick three examples from our ESM simulations (Fig. 3).

Arctic summer sea-ice is melting rapidly during the positive emission phases, resulting in a sea-ice-free Arctic during September (less than 1 million square km covered by sea-ice) in all B-simulations (Fig. S4). During the zero-emission phases summer sea-ice-cover follows the trends in SAT, leading to recovery until the SAT minimum is reached and a renewed decline thereafter. If CDR is applied, the amplified northern high-latitude cooling has a pronounced impact on the Arctic summer sea-ice extent (Fig. S4). Even in our low overshoot simulation, 72% of the area of the preindustrial Arctic summer sea-ice extent undergoes a thaw-refreeze-rethaw cycle within 200 years (methods, Fig. 3a). For comparison, this area is only 42% in the reference simulation and 85% (100%) in the $OS_0^{500}$ ($OS_0^{1000}$) overshoot. The retreat of sea-ice and changing environmental conditions will lead to gradual or rapid shifts in Arctic ecosystems through adaptation, migration, or extinction of species[27–30], and in general, such shifts are not easily reversible[31]. Therefore, a period of cooling after a warming-induced ecosystem shift might (instead of just reversing the shift) act as an additional stressor to the ecosystem and cause further loss of biodiversity. On the other hand, the retreat of Arctic sea-ice also brings economic opportunities in mining, gas and oil exploration[32], tourism[33], fishing[34], and shipping[35]. A pronounced but intermittent cooling with regrowth of sea-ice can be expected to have negative economic impacts, for example by closing newly opened Arctic shipping routes or by making large Arctic regions inaccessible for economic exploitation again. This will either hinder economic development (if the cooling is correctly predicted) or lead to stranded assets (if the cooling comes as a surprise).

Similar to summer sea-ice, the permafrost extent declines rapidly in response to warming and follows the SAT trend very closely afterward (Fig. S5). Consequently, permafrost thaws, refreezes, and thaws again over considerable areas in our overshoot scenarios (Fig. 3b). The extent of an area undergoing such cycles is 12, 17.7, and 20.9% of the preindustrial permafrost extent ($15.3 \times 10^6$ km² in our model) in the $OS_0^{250}$, $OS_0^{500}$, and $OS_0^{1000}$ simulations, respectively. Again, the area affected by thawing, refreezing, and thawing is significantly smaller in the reference simulation without CDR (7.8%). These results are consistent with a previous modeling study that suggests global scale permafrost degradation trajectories will highly depend on temperature, and the physical state of permafrost is reversible under global scale temperature reduction[36]. Given that nearly 5 million people inhabit permafrost areas[37], and different types of human infrastructure are built on permafrost (including transportation networks, mining projects, electrical power transmission lines, and water retaining structures as well as general buildings), permafrost thaw will entail considerable costs related to adapting infrastructure to the changing environmental conditions (e.g., ground subsidence, bearing capacity of the ground, uneven surface deformation)[38]. Several studies have provided quantitative assessments of potential economic impacts of permafrost thaw on various types of infrastructures[38–40]. Engineering solutions for permafrost environments are generally specialized, for example, ref. [39]. could not identify adaptation measures for permafrost thaw that were less expensive than complete infrastructure replacement. This might indicate that permafrost thaw-refreeze-rethaw cycles could significantly increase adaptation costs if permafrost thaw occurs twice during the nominal lifetime of human infrastructure.

The potential growth of farmed and wild Atlantic Salmon (*Salmo Salar*) is confined to a thermal window, ranging from optimal conditions around 14 °C to lethal conditions below 1.5 °C or above 19 °C[41]. In our overshoot simulations, the cycle of North Atlantic SST warming, cooling, and subsequent warming results

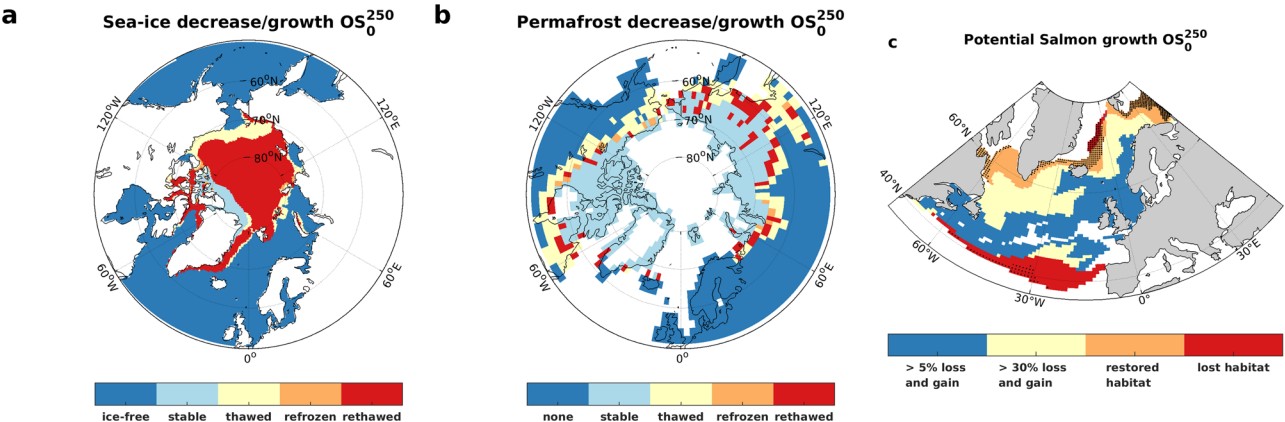

**Fig. 3 Impacts of warming-cooling-warming cycles.** Evolution of northern hemisphere **a** summer sea-ice, **b** permafrost, and **c** Atlantic salmon tolerable temperature range under the $OS_0^{250}$ overshoot simulation. In **a** and **b**, the light blue color indicates stable conditions, yellow color indicates permanent thaw, orange color indicates areas where thaw and refreezing occurred consecutively, and red areas indicate that a thaw-freeze-thaw cycle occurred (methods). In panel **c**, potential salmon growth rate loss and gain between 5 and 30% is indicated in blue, between 30 and 100% in yellow, and between lost and restored habitat in orange (methods). Habitat that is lost between SAT minimum and SAT recovery is marked in red. Stripeling indicates habitat gained during the final warming stage. Corresponding figures for the reference simulation and the $OS_0^{500}$ and $OS_0^{1000}$ overshoots can be found in the supplementary material (Figs. S6–S8).

in optimal growth conditions shifting northward, then southward, and finally northward again. As a consequence, in the $OS_0^{250}$ simulation, the potential growth rate undergoes alternating positive and negative changes with 40% of the preindustrial habitat being exposed to changes in the potential growth rate of 5–30% during each shift, 29% of the habitat to changes of 30–100%, and 26% of the habitat is lost and partly restored within shifts (Fig. 3c). In the reference run without overshoot, the corresponding changes in area are smaller and only 20% of the original habitat is lost and partly restored, while it is 36% in $OS_0^{500}$ and 40% in $OS_0^{1000}$. Such changes could have considerable economic consequences as farmed Atlantic Salmon is the most successful aquaculture species with the highest productivity in the northern hemisphere[42] and the exposed regions comprise areas with a high density of coastal aquaculture installations (Norway, Scotland, Faroe Island, and Canada)[42], as well as areas for emerging offshore aquaculture in Norway[43]. Ecological consequences for wild Atlantic Salmon could also be expected. Stocks of this migrating species have been declining over the last century due to anthropogenic stressors[44], and fluctuating temperature conditions along the natal rivers and migratory pathways will constitute an additional stressor and threaten the survival of an already vulnerable species.

**Robustness of cooling response.** How robust are the results of our simulations? To compare the behavior of our model with other ESMs we use two different simulations from the CMIP6 archive. The first one is a highly idealized simulation of CDR (1pctCO2-cdr) available through the carbon dioxide removal intercomparison project (CDRMIP)[17]. This simulation extends the standard experiment where atmospheric $CO_2$ is increased by 1% per year until quadrupling after 140 years, with a mirrored simulation where $CO_2$ is decreased by 1% per year until preindustrial concentrations have been restored after 280 years. In addition to NorESM2-LM, seven CMIP6 ESMs[45–51] have performed this simulation (Fig. 4a–c). Three models (CESM2, CNRM-ESM-2, and GFDL-ESM4) show initially a strong AMOC (>20 Sv at 40°N) and a strong decline (>10 Sv), similar to NorESM2-LM, which has the second strongest decline in absolute numbers (Fig. 4b). In the North Atlantic (between 47°N and 80°N), these models show a pronounced decline of surface

temperatures below the preindustrial level towards the end of the 1%-ramp down phase (Fig. 4c). One model (MIROC2-ES2L) that has a weaker initial AMOC strength and lesser decline also shows SAT below the preindustrial level in this region. The spatial cooling pattern is quite similar between the models with strong AMOC decline (Fig. S9). There is, however, also a dependence on how fast AMOC strength recovers from its minimum value, and on the intensity of global warming (i.e., on the transient climate response). For example, while the GFDL-ESM4 and CNRM-ESM2 models show a very similar decline in AMOC strength, recovery of AMOC is much faster in GFDL-ESM4. Therefore, the cooling is less pronounced in the latter model, since at the time the radiative forcing reaches preindustrial levels (year 280 of the simulation), AMOC strength has already completely recovered. NorESM2-LM has the lowest transient global warming of the eight ESMs, and therefore it initially cools faster than CESM2 which shows a similar decline in AMOC strength. We note that NorESM2 is based on CESM2 but implements a different ocean model. Therefore, the strong cooling seen in CESM2 indicates that this phenomenon is not a peculiarity of our ocean model.

The second simulation that we investigate is the ScenarioMIP[52] SSP5-3.4-overshoot scenario, for which output from nine models[46–50,53–55] (including NorESM2-LM) is available. In contrast to our idealized experiments and to the 1pctCO2-cdr simulation, this scenario has been created by an integrated assessment model and represents a plausible emission pathway with an unmitigated growth of emissions until 2040 and strong mitigation including negative emissions afterward. In the ensemble of nine ESMs, five models simulate a weaker and three models a stronger AMOC reduction by 2100 compared to NorESM2-LM (Fig. 4d, e). Four ESMs including our model show a decline of AMOC strength of more than 50% during the positive emission phase of this scenario. NorESM2-LM and two others of these models (CESM2-WACCM and MRI-ESM2) show a pronounced cooling below preindustrial levels in the North-Atlantic sector during the negative emission phase. Interestingly, these three models are the same models that exhibit rapid cooling events in the subpolar gyre region in some of the CMIP6 scenario simulation[56]. One model with pronounced AMOC reduction (GISS-E2-1-G) also shows a cooling trend in the North Atlantic sector, but the cooling is less pronounced (SAT does not fall

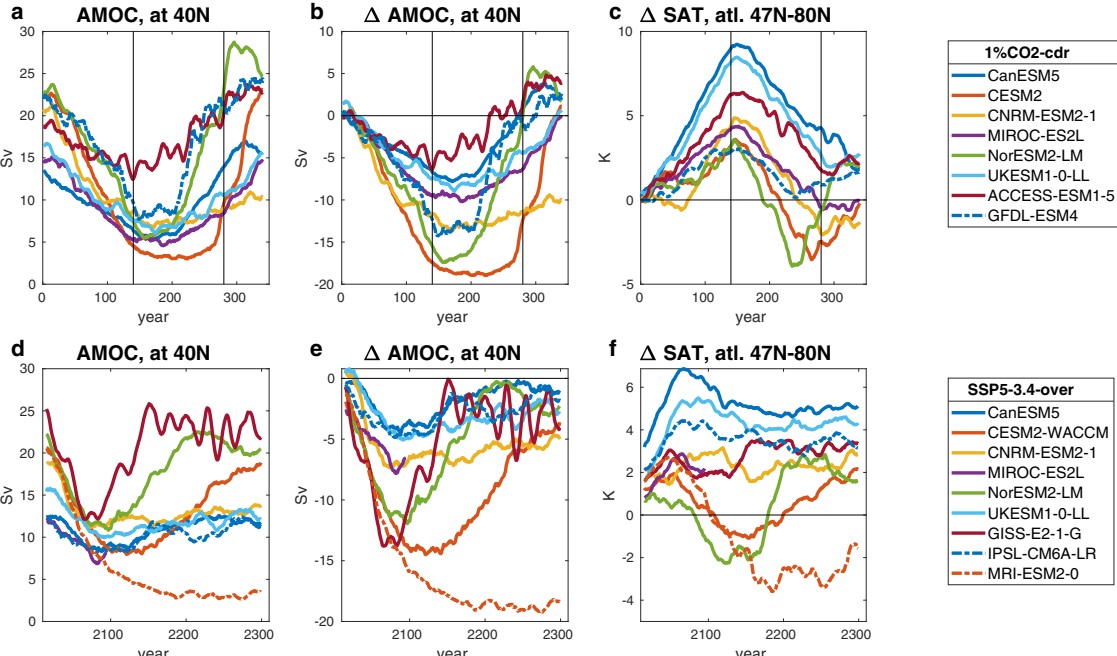

**Fig. 4 Northern high-latitude cooling in CDRMIP and ScenarioMIP simulations.** Results from the multi-model intercomparison projects CDRMIP and ScenarioMIP for **a**, **d** AMOC strength at 40°N, **b**, **e** change in AMOC strength relative to the preindustrial control simulation, and **c**, **f** change in SAT over the North Atlantic between 47 and 80°N. Panels **a–c** show results from the CDRMIP reversibility experiment (1pctCO2-cdr), where thin vertical lines mark the transition from 1% $CO_2$ increase per year to 1% decrease at year 140, and the transition from 1% $CO_2$ decrease to constant preindustrial $CO_2$ at year 280. Panels d-f show results for the ScenarioMIP SSP5-3.4-over simulation.

below preindustrial levels) due to a faster AMOC recovery in this model. In NorESM2-LM, SAT in the North Atlantic sector drops below preindustrial levels towards the end of the 21st century and reaches a minimum of 2° below preindustrial SAT around 2120. Thereafter, SAT stays at this low level for about 50 years before quickly bouncing up to more than 2° above preindustrial SAT as AMOC recovers. The magnitude and timescale of the SAT response in the North Atlantic sector between 47° and 80°N simulated by our model for the SSP5-3.4 scenario is consistent with the SAT response in our idealized $OS_0^{250}$ overshoot simulation (compare Fig. S3c). These results confirm that AMOC decline and northern high-latitude cooling in our model is within the range of results of other CMIP6 Earth system models and that the amplification of cooling in our idealized NorESM2-LM simulations is not significantly different compared to more realistic scenarios.

We note that some models show a strengthening of AMOC above preindustrial levels after the atmospheric $CO_2$ concentration has been returned to preindustrial levels in the CDRMIP 1pctCO2-cdr experiment (Fig. 4b). In our model, this leads to high-latitude warming following the cooling period (Fig. 4c). Such high-latitude warming has been described in earlier studies[15,16], which used a similar experimental setup, and acts to reinforce the warming-cooling-warming cycle described in this study. However, an acceleration of AMOC after a period of negative emissions seems to be a characteristic of simulations with extremely strong forcing such as the 1pctCO2-cdr experiment. There is no strengthening of AMOC above preindustrial levels in the SSP5-3.4 scenario simulation in any of the models, and we do not observe this in our idealized simulation either. In summary, these results demonstrate that a strong North Atlantic cooling under negative emissions seems to be a robust feature of Earth system models that show a strong and sustained AMOC decline in response to warming.

**Uncertainties**. Our study shows that phasing out $CO_2$ emissions could lead to a significant centennial cooling trend relative to peak temperatures north of 40°N if the preceding global warming has strongly weakened the Atlantic meridional overturning circulation. If net negative emissions are applied in a state of a strongly weakened AMOC, the temporary northern high-latitude cooling can be amplified, leading to a pronounced warming-cooling-warming pattern in overshoot scenarios. The ESM used in this study (NorESM2) shows a relatively strong decline of AMOC strength but is not an outlier compared to other state-of-the-art ESMs. Our simulations are idealized and the identified consequences for sea-ice, permafrost and ecosystems might not be directly transferable to more realistic mitigation scenarios. We do find, however, a similar pattern of northern high-latitude temperature fluctuations in a simulation of a plausible future scenario with net negative emissions. There is considerable uncertainty regarding the future fate of the AMOC, and this has been identified as one of the major sources of uncertainty in climate projections[23,24,57]. We are currently unable to assess how strongly and abruptly AMOC strength will change under climate warming and how fast the circulation might recover, but it has been suggested that ESMs, in general, tend to underestimate the possibility for abrupt and strong changes[58], which, in the case of AMOC, might be related to common biases[59]. Also, the melting of the Greenland ice sheet, which might further destabilize AMOC particularly on longer time scales[60] is not included in our and other state-of-the-art ESMs. Given the considerable impacts of the northern high-latitude cooling highlighted in our study, reducing the uncertainty of AMOC projections should be pursued with high priority.

## Methods
**NorESM simulations**. We use the Norwegian Earth System Model (NorESM2-LM)[12,13] in emission-driven configuration, that is, $CO_2$ is emitted into the atmosphere where it is advected with the atmospheric circulation and taken up or

released by the land biosphere and the ocean. $CO_2$ emissions are constructed following the protocol of the Zero-Emission Commitment Model Intercomparison Project (ZECMIP)[61] with total emissions of 1500, 1750, 2000, and 2500 Pg C over 100 years. Spatially, emissions are distributed uniformly over the sphere. Negative emissions for totals of 250, 500, and 1000 Pg C are constructed in the same way, but with a negative sign. No other forcing is varied in the model, that is, land use as well as non-$CO_2$ greenhouse gas and aerosol emissions and concentrations stay at preindustrial levels. For all simulations except the $B^{2000}$ and medium-sized over-shoots $OS_0^{500}$ and $OS_{100}^{500}$, we have run three ensemble members. We note that the climate sensitivity of NorESM2 is at the low end when compared to other CMIP6 models[62] and that most other models would show more warming for these levels of cumulative $CO_2$ emissions.

Cumulative emissions of 2500 Pg C are comparable to the $CO_2$ emissions of the Shared Socioeconomic Pathway (SSP) SSP5-8.5 ($2637 \pm 136$ Pg C)[63], while the 1500 Pg C emissions of the reference case correspond to SSP4-6.0 emissions ($1498 \pm 80$ Pg C). For comparison, emissions over the historical period from 1850 to 2019 are estimated at $650 \pm 65$ Pg C[1]. Scenarios that are consistent with limiting global warming to 1.5 °C until 2100 contain up to 330 Pg C negative emissions[2], although the uncertainties surrounding such estimates are large (e.g., ref. [3]). Also, the quoted 330 Pg C are gross negative emissions, and part of this carbon dioxide removal might be used to compensate for residual, difficult to mitigate positive emissions, and the amount of net negative emissions in these scenarios will be somewhat lower. Nevertheless, the amount of CDR in our low overshoot cases (250 Pg C) is clearly within the range of CDR discussed in the current scenario literature, while the amount of CDR applied in the medium overshoots (500 Pg C) is higher than this range, but still could be considered roughly consistent with IPCC scenarios given the longer timescale of our simulations. The amount of CDR applied in the high overshoot cases (1000 Pg C) most likely falls outside a feasible range. Here, we focus on the low and medium overshoot cases, but we include the results of the high overshoot simulations since they complement our understanding of the Earth system response to different levels of CDR.

**Sea-ice thaw-refreeze-rethaw cycles**. We define thaw of sea-ice if the September fractional ice-cover in a given grid cell over the first 11 years of the overshoot simulations changed by more than 50% relative to a 30-year preindustrial clima-tology. Refreeze is defined as an increase of September fractional ice-cover by 50% (or by an absolute increase of 0.3) during the sea-ice maximum in the negative emission phases relative to the first 11 years of the overshoots. Rethaw is defined as a thaw of previously refrozen grid points during the years 300–310 of the simulations when SAT and September sea-ice area become very similar in the reference and all short overshoot simulations.

**Permafrost thaw-refreeze-rethaw cycles**. We define permafrost grid points where the maximum active layer thickness is below 3 m. As for sea-ice, the thawed permafrost area is calculated for the first 11 years of the overshoot simulations relative to a 30-year preindustrial climatology. Refreeze is defined as the increase in permafrost area during the permafrost maximum in the negative emission phases relative to the first 11 years of the overshoots. Rethawed area is calculated for the years 300–310 of the simulations relative to the previous maximum.

**Potential salmon growth**. For calculation of potential salmon growth, we follow the thermal performance curve of ref. [41]. These authors specify the somatic growth of salmon (G, kg per month) as a piecewise linear function that is dependent on sea surface temperature (SST, in °C): $G(SST) = 0.0264 \times SST - 0.0396$ for SST <14 and $G(SST) = -0.066 \times SST + 1.254$ for SST $\geq 14$. At temperatures below the minimum temperature for growth (1.5 °C) and above the maximum temperature for growth (19 °C), G is set to zero to reflect the absence of growth. If for a given grid point, G is zero over the whole decade, we consider this grid point as non-habitable for Salmon.

## Data availability
The model data generated in this study are available through the Norwegian Research Data Archive/Bjerknes Climate Data Centre and can be accessed under https://doi.org/10.11582/2022.00012. The CMIP6 model output data used in this study are available from the Earth System Grid Federation (ESGF) servers under https://esgf.llnl.gov.

## Code availability
The source code of NorESM2 is available at https://doi.org/10.5281/zenodo.3905091

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

## Acknowledgements
All authors acknowledge funding from the Research Council of Norway (project IMPOSE, grant 294930) and the Bjerknes Centre for Climate Research (project LOES). J.S. has received funding from the European Union's Horizon 2020 research and inno-vation program under grant agreement No 820989 (project COMFORT). The work reflects only the authors' view; the European Commission and their executive agency are not responsible for any use that may be made of the information the work contains. Supercomputing and storage resources were provided by UNINETT Sigma2 (projects nn9708k/ns9708k).

## Author contributions
J.S. conceived the study, designed the model experiments, performed the model simu-lations, analyzed and interpreted the model data, and wrote the manuscript. A.A., N.G., and H.L. contributed to the model data analysis and interpretation and to the writing of the manuscript.

## Competing interests
The authors declare no competing interests.
