## [Peer Review File · Nature Communications]

General comments

This study analyses the impact of deliberate CO₂ removal in the atmosphere on the fate of the AMOC in sensitivity simulations from different climate models. It is shown that the AMOC response to such removal can strongly impact the regional recovery of climate in the northern high latitudes, with potential cooling whose ecological and economical impacts is poorly accounted for up to now when assessing such removal scenario.

This is a clear and well written article. The rationale is well-described and the idea proposed is quite new, since the concept of CO₂ removal has emerged as a serious possibility quite recently to my knowledge. What the paper is describing concerns the inertia of climate and potential non-linearities or hysteresis, which can seriously complicate the response of the climate system to mitigation effort in the future. In that respect, it is important to go beyond iconic global temperature target, which is an over-simplification of the issue related with climate changes. In that respect, I find this study of great utility.

Nevertheless, I have the feeling that the study is making quite strong hypotheses which minor the impacts and robustness of the results.

The main issue that I would like to highlight is the fact that the authors are here considering idealized experiments, and they apply their conclusions based on those experiments to realistic scenarios. This is a strong approximation. Indeed, realistic historical and future climatic projections are based on observed greenhouse gases emissions or concentration, and then consider various scenarios of emissions to assess potential future climate. In that respect, these simulations clearly account for the time scale of the emission pathways, which can play a crucial role for the climate. Here, the authors consider idealized emission scenarios, where CO₂ concentration increase very rapidly. This is certainly not realistic, and the time scales of the warming and of the climatic response, in particular responses of potential tipping elements like the AMOC, are therefore not well represented. This can be crucial, since time scales are clearly important in the response of the AMOC. This is a difficult issue to be accounted for, but I would like to urge the authors to try to evaluate if the effect they highlight here is also true in more realistic simulations (i.e. historical followed by ssp scenario including CO₂ removal).

I have noted an additional caveat that is partially tackled in the manuscript, but not sufficiently in my view: The evaluation of impacts of the cooling is quite speculative, and in particular, I find that the mean north of 40°N presented in Fig. 1 is not very meaningful. Indeed, there are very different regions concerned here, and I would like to see more precise estimates on the impact on land, sea ice and ocean areas. Indeed, most of the signal seems to be found over the ocean areas, which are not very densely populated.

Specific comments

Since no line numbering was provided, I have added it on the submitted word. I join the document with line numbers to help the authors in the review process:

- Line 24: this sentence is quite unclear. The models also show that above some threshold, there can be irreversible changes. I suggest to add “under given thresholds” after “reversible”.
- Line 63-68: the description of NorESM response in terms of AMOC as compared to CMIP6 is a bit fast. In which CMIP6 quantile is positioned this model in 2100? Is it a quite sensitive model in terms of AMOC response? The fact that “it is not an outlier” is not sufficient to know in my view.
- Line 88: it might be worth to further refine this rough estimate, separating land, sea ice and ocean covered regions. And also possibly, Europe and North America.
- Line 154: “we speculate”. Indeed, this is quite speculative. Do you have further insights to support this claim? The same is true concerning sea ice in the Arctic Ocean (line 156-159). I do not find the arguments very convincing as it stands.
- Line 174-176: I do not find the logical linkage of this sentence with the former estimate of the cost of permafrost thawing very clear. Why a thaw-refreeze-thaw will add additional cost? Can you be a bit more specific and convincing on this point?
- Line 206: “highly idealized”: I totally agree with this, and as mentioned in my main comment, this constitute a major caveat concerning the significance of this study, given that time scales of the emission and its link with the response of the climate system can be crucial. The authors need at least propose an evaluation of the implications of this, but I would like to urge them to go a bit further, and try to consider also a few simulations with more realistic settings.
- Table 1: It is unclear how this number have been computed. Please specify on which years the means and differences have been computed exactly (if only one year, it seems a bit weird...). Also, uncertainty estimates might be welcome, especially when ensemble are performed.
- There are a few references that might help to strengthen the argument of this paper:
 - Liu et al. (2017) showing that present-day climate models might be too stable, which might have considerable implications concerning the AMOC return that is found in most models and that maybe not realistic. This issue of AMOC reversibility might be further discussed at the end of the manuscript since it can be also be a crucial issue/impact, although not found in the models used here.
 - Bakker et al. (2016) showed that neglected Greenland ice sheet melting can add AMOC weakening, especially on the long-term, which is the time scale mainly assessed here. This is an additional process that is not considered here, but might feed the scope of the paper (that I can summarized as “greenhouse gas removal is an option, but it increases the risk of bringing the AMOC towards a collapse threshold”). A discussion of this missing process might be useful.
 - IPCC SROCC chapter 6 (Collins et al. 2019) provides an enlightening assessment of the various impacts of a substantial AMOC weakening. This is going far beyond the impacts noted here. In particular, the impacts on the ITCZ and

African monsoon from the densely inhabited West African region can be considered has a far more impactful aspect of AMOC response for human society (including migration pressure for western countries)

- NorESM has been found as a model showing decadal rapid cooling events in the North Atlantic (on top of the AMOC weakening) in a recent study (Swingedouw et al. 2021). This is something that might be worth to note when describing the model.

References

- Bakker, P. *et al.* 2016. Fate of the Atlantic Meridional Overturning Circulation: strong decline under continued warming and Greenland melting: AMOC projections for warming and GIS melt. *Geophys. Res. Lett.* 43: 12,252–12,260.
- Collins, M. *et al.* 2019. IPCC special report on the ocean and cryosphere in a changing climate. Chapter 6: Extremes, abrupt changes and managing risks.
- Liu, W., S. P. Xie, Z. Y. Liu and J. Zhu, 2017: Overlooked possibility of a collapsed Atlantic Meridional Overturning Circulation in warming climate. *Science Advances*, 3 (1), e1601666, doi:10.1126/sciadv.1601666.
- Swingedouw, D., *et al.* 2021: On the risk of abrupt changes in the North Atlantic subpolar gyre in CMIP6 models. *Annals of the NY Academy of Sciences*, published on line (<https://doi.org/10.1111/nyas.14659>)

REVIEWER COMMENTS

Reviewer #2 (Remarks to the Author):

Review of NCOMMS-21-27946-T

"Possibility for strong northern hemisphere high-latitude cooling under negative emissions" by Schwinger et al.

This paper reports a potential side effect of CO₂ removal on northern high latitude temperatures following the peak of atmospheric concentration from two overshoot scenarios based on the Norwegian earth system model. The work is of interest and directly policy relevant.

The key point is that, if CO₂ removal is applied too early, surface air temperature over the northern high latitudes will fall below the Paris agreement target temporarily before recovery. Although some other models may indicate similar behaviour, the uncertainty is large. This result is opposite to earlier research, e.g.

Wu, P., L. Jackson, A. Pardaens and N. Schaller, 2011: Extended warming of the northern high latitudes due to an overshoot of the Atlantic Meridional Overturning Circulation. *Geophys. Res. Lett.*, 38, L24704, doi:10.1029/2011GL049998.

Wu, P., J. Ridley, A. Pardaens, R. Leavine and J. Lowe, 2015: The reversibility of CO₂ induced climate change. *Clim. Dyn.*, 45(3-4), pp745-754. DOI: 10.1007/s00382-014-2302-6.

Wu et al. (2011) demonstrate a clear mechanism for the AMOC overshoot and extended warming of the northern high latitudes. Weakening of the AMOC causes a pile-up of warm/salty water in the subtropics due to reduced meridional transport. Following the recovery, these accumulated heat and salt drive the overshoot of the AMOC and associated warming.

The phenomenon shown in the Norwegian model may be another possibility but requires detailed analysis of the mechanisms and robustness. It is a "possibility" as the title rightly suggests. I would suggest some analysis of that model's control simulation and general performance of the model's AMOC as well as associated thermohaline processes. Democratic ensemble mean may be popular. Well documented individual model results have equal importance as ensemble mean may never occur in the real world.

Overall, research along this direction is noteworthy and helpful for decision makers. I would like the paper to strengthen on mechanisms and robustness of the main result before it can be considered for publication.

Response to comments from the editor and two reviewers

We would like to thank the editor and the two reviewers for the overall positive evaluation of our manuscript and for the constructive and helpful criticism. We have substantially revised our manuscript and addressed all points that have been raised as detailed in our point-by-point response below (original comments in grey, italic font). Verbatim changes or additions to the manuscript are highlighted in **this colour**.

1) Comments from the editor and summary of main changes

As you will see from the reports copied below, the reviewers raise important concerns. For publication in Nature Communications, we need to ensure that the importance of the timing of introduction of CO₂ removal technologies on AMOC overshoot needs to be shown in a robust and realistic way. Hence, it is crucial that the effects you discuss do not only exist in idealized model simulations, but also relate to processes in the real world, which the referees argue has not been demonstrated. In addition, they have concerns about the robustness of the findings. We find that these concerns limit the strength of the study, and therefore we ask you to address them with additional work. Without substantial revisions, we will be unlikely to send the paper back to review.

To strengthen the realism and robustness of our study, we demonstrate that a strong northern high latitude cooling is also found in more realistic scenarios. We use results from the SSP5-3.4-overshoot scenario that is part of CMIP6 ScenarioMIP. This scenario has been created by an integrated assessment model and represents a plausible emission pathway with unmitigated growth of emissions until 2040 and strong mitigation including negative emissions afterwards. Results from this scenario simulation confirm that there is a considerable northern high latitude cooling not only in NorESM2-LM but also in other CMIP6 models that show a strong decline in AMOC under the positive emission phase of this scenario (see Fig. R1 d-f below). In the NorESM2-LM simulation of this scenario, surface air temperature (SAT) in the North Atlantic sector drops below pre-industrial values towards the end of the 21st century and reaches a minimum of 2° below pre-industrial SAT around 2120. Thereafter, SAT stays at this low level for about 50 years before quickly bouncing up to more than 2° above pre-industrial SAT as AMOC recovers. The magnitude of the cooling and subsequent warming in this scenario is similar to our idealized “low” overshoot simulation. This clearly demonstrates that our idealized simulations do not behave significantly differently from more realistic scenarios as far as the AMOC decline and northern high latitude cooling is concerned. The results also further underscore that the cooling seems to be a robust feature of Earth system models that show a strong AMOC decline.

We would like to emphasize that, although our simulations are idealized, they are closer to a realistic scenario than other, commonly used, idealized overshoot experiments. For example, the CMIP6 CDR-reversibility experiment described by Keller et al. (2018) starts with the standard CMIP 1% CO₂ increase experiment followed by a 1% decrease of CO₂ until pre-industrial CO₂ is restored, which results in an extremely abrupt change from implied positive to negative emissions. The simulations analysed in Wu et al. (2011, 2015) mentioned by reviewer #2 are of this type. Our simulations have a period of increasing

emissions followed by a period of decreasing emissions, and the positive and negative emission phases are smoothly joined. Moreover, CO₂ concentrations in our experiments are not returned to preindustrial levels, but to a level compatible with keeping global temperature increase well below 2 degrees.

Reviewer #2 got the impression that our model behaves opposite than the ESMs studied by Wu et al. (2011, 2015), but this is not the case. Our model behaves very similarly to these ESMs when looking at comparable idealized experiments. In the CDR-reversibility experiment (Fig. R1 a-c), our model shows the same type of AMOC overshoot as observed in Wu et al. (2011, 2015): After pre-industrial CO₂ has been restored, AMOC “overshoots” the pre-industrial level, and SAT exhibits a prolonged warming in the North Atlantic sector. This warming comes *after* the pronounced cooling during the phase where atmospheric CO₂ levels are reduced (please see the response to reviewer #2 for more details). We note that also other ESMs show this “AMOC overshoot” behaviour (4 out of 8 models that have provided the necessary data) after the atmospheric CO₂ concentrations have returned to pre-industrial levels.

Fig R1: New Fig. 4 of the revised manuscript, copied here for convenience. Results from the multi-model intercomparison projects CDRMIP and ScenarioMIP for (a,d) AMOC strength at 40°N, (b,e) change in AMOC strength relative to the pre-industrial control simulation, and (c,f) change in SAT over the North Atlantic between 47 and 80°N. Panels a-c show results from the CDRMIP reversibility experiment, where thin vertical lines mark the transition from 1% CO₂ increase per year to 1% decrease at year 140, and the transition from 1% CO₂ decrease to constant pre-industrial CO₂ at year 280. Panels d-f show results for the ScenarioMIP SSP5-3.4-over simulation.

In the more realistic SSP5-3.4-over scenario simulation, the AMOC does not overshoot pre-industrial strength after its recovery in our model (and it does not do so in any of the other models). This indicates that there are indeed (as suggested by reviewer #1) important differences between very idealized simulations and more realistic scenarios. This is particularly true in idealized simulations with unrealistically large and fast changes in forcing (such as the CDR-reversibility experiment and the experiments used in Wu et al. (2011, 2015)). In contrast, our idealized simulations seem to be realistic enough to show a qualitatively similar AMOC behaviour as the SSP5-3.4-over scenario simulation.

In summary, our revision including additional analyses addresses the reviewers' concerns by demonstrating that

- Our results still hold for more realistic scenarios, evidenced with the CMIP6 SSP5-3.4-over scenario simulation
- Our model agrees with other models in previous studies showing a very similar behaviour if we consider the same type of idealized experiment.

Summary of main changes to the manuscript

- We substantially extended the subsection "Robustness of cooling response" to include a discussion of the SSP5-3.4 overshoot scenario and a comparison with other ESMs that have performed this scenario simulation. This also includes a short clarification of "AMOC overshooting" and high latitude warming in the CDRMIP reversibility experiment and in Wu et al. (2011,2015) to avoid any confusion around seemingly opposite model behaviour compared to previous studies.
- We have added text at the beginning of the "Results" section describing better how our model performs with respect to observation based estimates of AMOC strength, and how it performs compared to other CMIP6 models.
- We have extended our analysis and text in the subsection "Impacts and challenges for adaptation" concerning the evaluation of possible impacts and backed up our reasoning with more references to existing literature. We have refined the analysis of temperature response to the different regions (land, broken down into continents and ocean). In the abstract we formulate the sentence "We show that this behaviour would pose economic and ecological risks..." a bit more careful by rewording to "We show that this behaviour may cause economic and ecological impacts...".

Please see our responses to the individual reviewers' comments below for details.

2) Response to reviewer #1

This study analyses the impact of deliberate CO₂ removal in the atmosphere on the fate of the AMOC in sensitivity simulations from different climate models. It is shown that the AMOC response to such removal can strongly impact the regional recovery of climate in the northern high latitudes, with potential cooling whose ecological and economical impacts is poorly accounted for up to now when assessing such removal scenario.

This is a clear and well written article. The rationale is well-described and the idea proposed is quite new, since the concept of CO₂ removal has emerged as a serious possibility quite recently to my knowledge. What the paper is describing concerns the inertia of climate and potential non-linearities or hysteresis, which can seriously complicate the response of the climate system to mitigation effort in the future. In that respect, it is important to go beyond iconic global temperature target, which is an oversimplification of the issue related with climate changes. In that respect, I find this study of great utility.

Thank you for this positive overall evaluation.

Nevertheless, I have the feeling that the study is making quite strong hypotheses which minor the impacts and robustness of the results. The main issue that I would like to highlight is the fact that the authors are here considering idealized experiments, and they apply their conclusions based on those experiments to realistic scenarios. This is a strong approximation. Indeed, realistic historical and future climatic projections are based on observed greenhouse gases emissions or concentration, and then consider various scenarios of emissions to assess potential future climate. In that respect, these simulations clearly account for the time scale of the emission pathways, which can play a crucial role for the climate. Here, the authors consider idealized emission scenarios, where CO₂ concentration increase very rapidly. This is certainly not realistic, and the time scales of the warming and of the climatic response, in particular responses of potential tipping elements like the AMOC, are therefore not well represented. This can be crucial, since time scales are clearly important in the response of the AMOC. This is a difficult issue to be accounted for, but I would like to urge the authors to try to evaluate if the effect they highlight here is also true in more realistic simulations (i.e. historical followed by ssp scenario including CO₂ removal).

In the revised version of our manuscript, we demonstrate that our model also simulates a strong northern hemisphere cooling in the CMIP6 SSP5-3.4-over scenario. Such cooling is also shown in two other ESMs in this scenario. Please also see our general response above.

Changes to the manuscript: We have substantially extended the subsection “Robustness of cooling response” to include a discussion of the SSP5-3.4 overshoot scenario and a comparison with other ESMs that have performed this scenario simulation:

“The second simulation that we investigate is the ScenarioMIP⁵³ SSP5-3.4-overshoot scenario, for which output from nine models^{47–51,54–56} (including NorESM2-LM) is available. In contrast to our idealized experiments and to the 1pctco2-cdr simulation, this scenario has been created by an integrated assessment model and represents a plausible emission pathway with an unmitigated growth of emissions until 2040 and strong mitigation including negative emissions afterwards. In the ensemble of

nine ESMs five models simulate a weaker and three models a stronger AMOC reduction by 2100 compared to NorESM2-LM (Fig. 4d-e). Four ESMs including our model show a decline of AMOC strength of more than 50% during the positive emission phase of this scenario. NorESM2-LM and two other of these models (CESM2-WACCM and MRI-ESM2) show a pronounced cooling below pre-industrial levels in the North-Atlantic sector during the negative emission phase. Interestingly, these three models are the same models that exhibit rapid cooling events in the subpolar gyre region in some of the CMIP6 scenario simulation⁵⁷. One model with pronounced AMOC reduction (GISS-E2-1-G) also shows a cooling trend in the North Atlantic sector, but the cooling is less pronounced (SAT does not fall below pre-industrial levels) due to a faster AMOC recovery in this model. In NorESM2-LM, SAT in the North Atlantic sector drops below pre-industrial levels towards the end of the 21st century and reaches a minimum of 2° below pre-industrial SAT around 2120. Thereafter, SAT stays at this low level for about 50 years before quickly bouncing up to more than 2° above pre-industrial SAT as AMOC recovers. The magnitude and time scale of the SAT response in the North Atlantic sector between 47° and 80°N simulated by our model for the SSP5-3.4 scenario is consistent with the SAT response in our idealized OS_0^{250} overshoot simulation (compare Fig. S3c). These results confirm that AMOC decline and northern high latitude cooling in our model is within the range of results of other CMIP6 Earth system models, and that the amplification of cooling in our idealized NorESM2-LM simulations is not significantly different compared to more realistic scenarios.”

I have noted an additional caveat that is partially tackled in the manuscript, but not sufficiently in my view: The evaluation of impacts of the cooling is quite speculative, and in particular, I find that the mean north of 40°N presented in Fig. 1 is not very meaningful. Indeed, there are very different regions concerned here, and I would like to see more precise estimates on the impact on land, sea ice and ocean areas. Indeed, most of the signal seems to be found over the ocean areas, which are not very densely populated.

We have extended our analysis and text concerning the evaluation of impacts and backed up our reasoning with more references to existing literature (see also our replies to the specific comments below). We have refined the analysis of temperature response to the different regions as suggested by the reviewer and included a new figure (similar to Fig 1h, but showing SAT over the ocean, over land, as well as over Europe, Asia, and North America) in the supplement, copied below as Fig. R2 for convenience. Absolute temperature anomalies over the ocean show lower temperatures than over land, particularly over the North Atlantic (as expected). European surface temperature anomalies are lowest of all continents due to the proximity and downwind location relative to the source of the anomaly. However, our analysis also shows that the additional cooling in the overshoots compared to the reference simulation without overshoot is generally very similar for the different regions (Fig. R2d-f). There is a tendency that land masses located further downwind from the source of the anomaly in the North Atlantic experience less extreme additional cooling during the overshoot. This effect can clearly be seen in the highest overshoot (Fig. R2a,d), but it is less pronounced in the low overshoot case. This indicates that the regional differences with respect to the additional cooling during an overshoot in our model are relatively small and that the cooling is spatially very coherent over the region north of 40°N.

Changes to the manuscript: We have added the following text to the “Results” section:

“Further analysis of the temperature response north of 40°N indicates that absolute temperature anomalies (relative to pre-industrial temperature) in the overshoot simulations remain smaller over the ocean than over land, with a particularly strong cooling over the North Atlantic (Fig. S3a-c), as expected from the strong reduction of AMOC strength. Absolute surface temperature anomalies in Europe are smallest of all continents due to the proximity and downwind location relative to the North-Atlantic. However, our analysis also shows that the additional cooling in the overshoot simulations relative to the reference (i.e. the simulation without overshoot) is regionally very similar north of 40°N (Fig. S3d-f). There is a tendency that land masses located further downwind from the source of the anomaly in the North Atlantic experience less additional cooling during the overshoot, i.e. the additional cooling is weaker in North America than in Asia and weaker in Asia than in Europe. This effect can clearly be seen for the highest overshoot, but it is less pronounced in the low overshoot case (Fig. S3a,d). This indicates that the amplified cooling north of 40°N during an overshoot in our model is spatially very coherent with only minor regional differences.”

Fig R2: New Fig. S3 of the revised supplementary material, copied here for convenience. Difference in (a-c) surface air temperature north of 40°N relative to the pre-industrial control simulation for the three “short” overshoots as indicated in the panel title, and (d-f) same as panels a-c but relative to the reference simulation (simulation σ^{1500} without overshoot). Shown are temperature differences over the ocean (blue lines), the North Atlantic between 47° and 80°N (light blue lines), over land (red lines), Europe (yellow lines), Asia (purple lines), and North America (green lines).

Since no line numbering was provided, I have added it on the submitted word. I join the document with line numbers to help the authors in the review process:

- *Line 24: this sentence is quite unclear. The models also show that above some threshold, there can be irreversible changes. I suggest to add “under given thresholds” after “reversible”.*

We have added “if certain thresholds are not crossed” after reversibel.

- *Line 63-68: the description of NorESM response in terms of AMOC as compared to CMIP6 is a bit fast. In which CMIP6 quantile is positioned this model in 2100? Is it a quite sensitive model in terms of AMOC response? The fact that “it is not an outlier” is not sufficient to know in my view.*

In our revised manuscript, we have extended the description of how the simulated contemporary AMOC in NorESM2-LM compares to observation based estimates and to the CMIP ensemble. We also have refined our description of the strength of AMOC decline in NorESM2-LM relative to the CMIP6 model ensemble. However, it is not easily possible to give a ranking for our model relative to “the” CMIP6 ensemble, since this depends on the forcing and different models have run different CMIP6 experiments. In the work of Bellomo et al. (2021) our model is among the 4 ESMs with the strongest decline. Weijer et al. (2021) show that our model has the strongest AMOC decline in the SSP1-2.6, SSP2-4.5, and SSP5-8.5 scenarios, but our own analysis of SSP5-3.4 shows that there are three models with a stronger AMOC decline in 2100 for this scenario. We therefore refrain from specifying a “CMIP6 quantile”, but rather discuss the AMOC decline in our model in the 1pctCO₂-cdr simulation and in the SSP5-3.4 overshoot scenario in more detail in subsection “Robustness of the cooling response” in the revised manuscript.

Changes to the manuscript:

In the first paragraph of the “Results” section, we replaced the sentence “We note that our model shows a slightly stronger than observed AMOC and a pronounced decline, but it is well within the range of other recent Earth system models (ESMs)” by the following text:

“We note that our model shows a slightly stronger than observed AMOC and a pronounced decline, but it is well within the range of other recent Earth system models (ESMs)^{17,18}. the simulated contemporary AMOC strength at 26°N in our model is 21 Sv and compares reasonably well to the observation based estimate of 17.4 Sv²⁰. NorESM2-LM is among the 4 CMIP6 models (disregarding one outlier model) with the highest contemporary overturning strength (CMIP6 range: 9.6 to 23 Sv; CMIP6 mean: 17.7 Sv)²⁰. The model simulates the depth of the AMOC maximum at 0.9 km slightly better than the CMIP6 model mean (0.84 km), but still shallower than the observation based estimate (1.04 km)²⁰. Our model generally shows a pronounced decline of AMOC strength with climate change. We discuss this point further below, where we compare the AMOC weakening in our model to other CMIP6 models.”

In the subsection “Robustness of the cooling response” we added a discussion of AMOC response in our and other CMIP6 models in the 1pctCO2-cdr simulation:

“Three models (CESM2, CNRM-ESM-2, and GFDL-ESM4) show initially a strong AMOC (>20 Sv at 40°N) and a strong decline (>10 Sv), similar to NorESM2-LM, which has the second strongest decline in absolute numbers (Fig. 4b).”

and the SSP5-3.4-over scenario:

“In the ensemble of nine ESMs five models simulate a weaker and three models a stronger AMOC reduction by 2100 compared to NorESM2-LM (Fig. 4d-e). Four ESMs including our model show a decline of AMOC strength of more than 50% during the positive emission phase of this scenario.”

- *Line 88: it might be worth to further refine this rough estimate, separating land, sea ice and ocean covered regions. And also possibly, Europe and North America.*

We have refined our analysis of surface temperature response, please see our response above.

- *Line 154: “we speculate”. Indeed, this is quite speculative. Do you have further insights to support this claim? The same is true concerning sea ice in the Arctic Ocean (line 156-159). I do not find the arguments very convincing as it stands.*

It is widely accepted that global warming is a threat to ecosystem resilience and biodiversity. Global warming will lead to rapid or gradual shifts in ecosystems through adaptation, migration, or extinction of species (e.g., Huntington et al. 2020; Wernberg et al. 2016; Rasher et al. 2020). Our reasoning here is that this must be true not only for climate change related to global warming, but more generally for changing environmental conditions, specifically also for a cooling after a period of warming (particularly for a cooling below pre-industrial levels). Arctic ecosystems are possibly already shifting towards a new state (Huntington et al. 2020), but in general such shifts are not easily reversible (e.g., Wernberg et al. 2016). A period of cooling after an ecosystem shift might therefore (instead of just reversing the shift) act as an additional stressor to the ecosystem and cause additional loss of biodiversity.

Regarding economic opportunities, if we take shipping as an example, the opening of new, considerably shorter, seaways between Europe/North America and East Asia is usually seen as an economic opportunity, since it might reduce the cost for transportation of goods. In our simulations, the re-growth of sea ice would close these new routes again. The same applies for fixed infrastructure used for mining or oil/gas exploration, which cannot expand northwards if sea ice is expected to re-grow (within the lifetime of this infrastructure).

Changes to the manuscript:

In the revised manuscript we explain our reasoning better by replacing the text “The retreat of sea ice entails most likely severe consequences for Arctic marine ecosystems (e.g. refs. ^{28,29}) and

we speculate that a thaw-refreeze-rewarm cycle might act as an additional stressor and overwhelm the adaptive capacity of fragile Arctic ecosystems” by:

“The retreat of sea ice and changing environmental conditions will lead to gradual or rapid shifts in Arctic ecosystems through adaptation, migration, or extinction of species^{28–31}, and in general such shifts are not easily reversible³². Therefore, a period of cooling after a warming induced ecosystem shift might (instead of just reversing the shift) act as an additional stressor to the ecosystem and cause further loss of biodiversity.”

We better explain our reasoning economic opportunities as follows:

“A pronounced but intermittent cooling with a re-growth of sea-ice can be expected to have negative economic impacts, for example by closing newly opened Arctic shipping routes or by making large Arctic regions inaccessible for economic exploitation again. This will either hinder economic development (if the cooling is correctly predicted) or lead to stranded assets (if the cooling comes as a surprise).”

- *Line 174-176: I do not find the logical linkage of this sentence with the former estimate of the cost of permafrost thawing very clear. Why a thaw-refreeze-thaw will add additional cost? Can you be a bit more specific and convincing on this point?*

In our revised manuscript we have added references to additional literature (Streletskiy et al. 2019; Hjort et al. 2018), have expanded our text, and formulated a bit more carefully, as follows:

Changes to the manuscript:

“...permafrost thaw will entail considerable costs related to adapting infrastructure to the changing environmental conditions (e.g., ground subsidence, bearing capacity of the ground, uneven surface deformation)³¹. Several studies have provided quantitative assessments of potential economic impacts of permafrost thaw on various types of infrastructures^{31–33}. Engineering solutions for permafrost environments are generally specialized, for example, ref.³² could not identify adaptation measures for permafrost thaw that were less expensive than complete infrastructure replacement. This might indicate that permafrost thaw-refreeze-rewarm cycles could significantly increase adaptation costs if permafrost thaw occurs twice during the nominal lifetime of human infrastructure.”

- *Line 206: “highly idealized”: I totally agree with this, and as mentioned in my main comment, this constitute a major caveat concerning the significance of this study, given that time scales of the emission and its link with the response of the climate system can be crucial. The authors need at least propose an evaluation of the implications of this, but I would like to urge them to go a bit further, and try to consider also a few simulations with more realistic settings.*

In our revised manuscript we analyze the CMIP6 SSP5-3.4-over scenario (please see our general response above). This scenario has been created by an integrated assessment model and

represents a plausible emission pathway with unmitigated growth of emissions until 2040 and strong mitigation including negative emissions afterwards.

- *Table 1: It is unclear how this number have been computed. Please specify on which years the means and differences have been computed exactly (if only one year, it seems a bit weird...). Also, uncertainty estimates might be welcome, especially when ensemble are performed.*

These numbers refer to the ensemble means of the 11-year running mean temperature or AMOC strength. We have added this information to the table caption. We have further added the ensemble range as an estimate of uncertainty.

- *There are a few references that might help to strengthen the argument of this paper:*
 - o *Liu et al. (2017) showing that present-day climate models might be too stable, which might have considerable implications concerning the AMOC return that is found in most models and that maybe not realistic. This issue of AMOC reversibility might be further discussed at the end of the manuscript since it can be also be a crucial issue/impact, although not found in the models used here.*

o Bakker et al. (2016) showed that neglected Greenland ice sheet melting can add AMOC weakening, especially on the long-term, which is the time scale mainly assessed here. This is an additional process that is not considered here, but might feed the scope of the paper (that I can summarized as “greenhouse gas removal is an option, but it increases the risk of bringing the AMOC towards a collapse threshold”). A discussion of this missing process might be useful.

Thank you very much for pointing us towards these papers. We have included the two above papers in the “Conclusions” section as follows.

Changes to the manuscript: We have extended the sentence “We are currently unable to assess how strongly and abruptly AMOC strength will change under climate warming and how fast the circulation might recover, but it has been suggested that ESMs in general tend to underestimate the possibility for abrupt and strong changes⁵⁷.” to

“We are currently unable to assess how strongly and abruptly AMOC strength will change under climate warming and how fast the circulation might recover, but it has been suggested that ESMs in general tend to underestimate the possibility for abrupt and strong changes⁵⁹, which, in the case of AMOC, might be related to common biase⁶⁰. Also, the melting of the Greenland ice sheet, which might further destabilize AMOC particularly on longer time scales⁶¹ is not included in state of the art ESMs.”

o IPCC SROCC chapter 6 (Collins et al. 2019) provides an enlightening assessment of the various impacts of a substantial AMOC weakening. This is going far beyond the impacts noted here. In particular, the impacts on the ITCZ and African monsoon from the densely inhabited West African

region can be considered has a far more impactful aspect of AMOC response for human society (including migration pressure for western countries)

Thank you, we have included a short discussion of impacts of an AMOC weakening on the low latitudes.

Changes to the manuscript: In the subsection “Impacts and challenges for adaptation”, we add the text

“An AMOC slowdown or collapse is thought to have severe impacts globally. Most notably, a southward shift of tropical rainfall and an increase of droughts in the Amazon and Sahel regions²⁶ would affect a large and vulnerable population. Here, we focus on the high latitude temperature fluctuations seen in our overshoot simulations noting that the AMOC slowdown itself is in fact mitigated by application of CDR in our model (i.e. AMOC recovery happens faster with CDR than without).”

o NorESM has been found as a model showing decadal rapid cooling events in the North Atlantic (on top of the AMOC weakening) in a recent study (Swingedouw et al. 2021). This is something that might be worth to note when describing the model.

Thank you indeed for making us aware of this paper. Interestingly, the 3 models with the abrupt cooling events are also those showing a strong cooling response in the SSP5-3.4 simulation.

Changes to the Manuscript: In the subsection “Robustness of cooling response”, we add the following sentence when discussing the 3 models that show a strong cooling under the SSP5-3.4-over scenario:

“Interestingly, these three models are the same models that exhibit rapid cooling events in the subpolar gyre region in some of the CMIP6 scenario simulations⁵⁷.”

References

Bakker, P. et al. 2016. Fate of the Atlantic Meridional Overturning Circulation: strong decline under continued warming and Greenland melting: AMOC projections for warming and GIS melt. Geophys. Res. Lett. 43: 12,252–12,260.

Collins, M. et al. 2019. IPCC special report on the ocean and cryosphere in a changing climate. Chapter 6: Extremes, abrupt changes and managing risks.

Liu, W., S. P. Xie, Z. Y. Liu and J. Zhu, 2017: Overlooked possibility of a collapsed Atlantic Meridional Overturning Circulation in warming climate. Science Advances, 3 (1), e1601666, doi:10.1126/sciadv.1601666.

Swingedouw, D., et al. 2021: On the risk of abrupt changes in the North Atlantic subpolar gyre in CMIP6 models. Annals of the NY Academy of Sciences, published on line (<https://doi.org/10.1111/nyas.14659>)

3) Response to Reviewer #2

This paper reports a potential side effect of CO₂ removal on northern high latitude temperatures following the peak of atmospheric concentration from two overshoot scenarios based on the Norwegian earth system model. The work is of interest and directly policy relevant.

Thank you for this positive overall evaluation.

The key point is that, if CO₂ removal is applied too early, surface air temperature over the northern high latitudes will fall below the Paris agreement target temporally before recovery. Although some other models may indicate similar behaviour, the uncertainty is large. This result is opposite to earlier research, e.g.

*Wu, P., L. Jackson, A. Pardaens and N. Schaller, 2011: Extended warming of the northern high latitudes due to an overshoot of the Atlantic Meridional Overturning Circulation. *Geophys. Res. Lett.*, 38, L24704, doi:10.1029/2011GL049998.*

*Wu, P., J. Ridley, A. Pardaens, R. Leavine and J. Lowe, 2015: The reversibility of CO₂ induced climate change. *Clim. Dyn.*, 45(3-4), pp745-754. DOI: 10.1007/s00382-014-2302-6.*

Wu et al. (2011) demonstrate a clear mechanism for the AMOC overshoot and extended warming of the northern high latitudes. Weakening of the AMOC causes a pile-up of warm/salty water in the subtropics due to reduced meridional transport. Following the recovery, these accumulated heat and salt drive the overshoot of the AMOC and associated warming.

The phenomenon shown in the Norwegian model may be another possibility but requires detailed analysis of the mechanisms and robustness. It is a “possibility” as the title rightly suggests. I would suggest some analysis of that model’s control simulation and general performance of the model’s AMOC as well as associated thermohaline processes. Democratic ensemble mean may be popular. Well documented individual model results have equal importance as ensemble mean may never occur in the real world.

As already mentioned in our general response above, our model does not behave differently from the models studied in Wu et al. (2011, 2015). The pile-up of warm and salty water in the subtropics, and AMOC overshooting causing high latitude warming on recovery is very similar in our model if we look at a comparable experiment, for example, the 1pctCO₂-cdr experiment. This experiment was already discussed in the original manuscript, and our model shows a pronounced AMOC “overshoot” and northern high latitude warming *after* pre-industrial CO₂ concentrations have been restored (please see Fig. R1 a-c above for AMOC and SAT time series). In Fig. R3 (see below) we show a Hovmöller diagram of Atlantic zonal mean density and its decomposition into salinity and temperature contribution for the 1pctCO₂-cdr simulation, similar to Fig. 3 of Wu et al. 2011. Please note that Fig. 3 in Wu et al. 2011 is

based on a slightly different experiment, in which the rate of increase until 4xCO₂ and subsequent decrease is larger (2% compared to 1% for the 1pctCO₂-cdr simulation). Therefore, the density anomalies in our model in Fig. R3 below appear to be more stretched out in time compared to the results of Wu et al. (2011). Otherwise this figure confirms that there is no significantly different behaviour in our model. The high latitude freshening in our model is larger, consistent with the larger AMOC sensitivity in our model compared to Wu et al. (2011). The main difference is the positive temperature contribution around 60°N to the density anomaly. This is due to the high latitude cooling as a consequence of AMOC slowdown in our model. Around year 280, the positive contribution of temperature to the density anomaly switches sign, since warm and salty waters are flushed into the high latitudes upon AMOC recovery.

Figure R3: Hovmöller diagram of Atlantic zonal mean upper ocean (top 800 m depth) density (left panel), and the contributions of temperature (middle panel) and salinity (right panel) to density anomalies in the 1pctCO₂-cdr simulation with NorESM2-LM.

Changes to the manuscript: We have added a paragraph in the subsection “Robustness of cooling response” that describes the “AMOC overshooting” behaviour in our model to better put our results into perspective with the existing literature:

“We note that some models show a strengthening of AMOC above pre-industrial levels after the atmospheric CO₂ concentration has been returned to pre-industrial levels in the CDRMIP 1pctCO₂-cdr experiment (Fig. 4b). In our model, this leads to a high latitude warming following the cooling period (Fig. 4c). Such high latitude warming has been described in earlier studies^{16,17}, which used a similar experimental setup, and acts to reinforce the warming-cooling-warming cycle described in this study. However, an acceleration of AMOC after a period of negative emissions seems to be a characteristic of simulations with extremely strong forcing such as the 1pctCO₂-cdr experiment. There is no

strengthening of AMOC above pre-industrial levels in the SSP5-3.4 scenario simulation in any of the models, and we do not observe this in our idealized simulation either.”

Overall, research along this direction is noteworthy and helpful for decision makers. I would like the paper to strengthen on mechanisms and robustness of the main result before it can be considered for publication.

We have substantially expanded our manuscript to assess the robustness of our results. We have added an analysis of a more realistic scenario that includes CDR from CMIP6 ScenarioMIP (SSP5-3.4-over). In this scenario simulation, our model shows a very similar response (strong northern high latitude cooling) under negative emissions compared to our idealized experiments. We have also expanded the description of our model to provide a more detailed description of how contemporary AMOC strength compares to observation based estimates and other CMIP6 models. We would like to stress again that there is no new or unusual mechanism in our model. The AMOC weakening and recovery itself follow the well known mechanism, and although our model tends to be on the sensitive side compared to other models, it is well within the range of behaviour found for other state-of-the-art Earth system models.

References

- Hjort, J. *et al.* Degrading permafrost puts Arctic infrastructure at risk by mid-century. *Nat. Commun.* **9**, 5147 (2018).
- Huntington, H. P. *et al.* Evidence suggests potential transformation of the Pacific Arctic ecosystem is underway. *Nat. Clim. Change* **10**, 342–348 (2020).
- Rasher, D. B. *et al.* Keystone predators govern the pathway and pace of climate impacts in a subarctic marine ecosystem. *Science* **369**, 1351–1354 (2020).
- Streletskiy, D. A., Suter, L. J., Shiklomanov, N. I., Porfiriev, B. N. & Eliseev, D. O. Assessment of climate change impacts on buildings, structures and infrastructure in the Russian regions on permafrost. *Environ. Res. Lett.* **14**, 025003 (2019).
- Wernberg, T. *et al.* Climate-driven regime shift of a temperate marine ecosystem. *Science* **353**, 169–172 (2016).

REVIEWERS' COMMENTS

Reviewer #1 (Remarks to the Author):

This is my second review of this paper.

The authors have correctly accounted for the suggestions I have provided in the first round. The inclusion of the intercomparison model simulations with more realistic settings are indeed still showing the effect they highlight in the idealized one. Therefore, this argument is convincing as well as the effect that it is robust in other models. Thus, I think the paper is now suitable for publication.

I have only minor comments that the authors might find interesting to integrate in their manuscript to improve clarity and impacts.

In the introduction, regarding the discussion of cumulative carbon emissions (lines 34 to 51), the authors do not mention that we've already emitted about 2500 Pg C according to Fig. 10 from SPM of the IPCC AR6 (2021). This is an element that might be of interest for the reader to know also, in order to better assess the idealized experiments. A large amount of the climatic impacts of those past emissions might have been damped by anthropogenic aerosols.

When discussing about the impacts (from line 150), the authors might be interested in this paper showing that a reversal of warming over Europe might also strongly affect multi-decadal agriculture choices like viticulture (Sgubin et al. 2019). Nevertheless, discussing this might possibly not fit with how the discussion is presented at the moment, so I just mention it for their knowledge.

Reference:

Sgubin G. , Swingedouw D., Garcia de Cortazar-Atauri I., Ollat N. and van Leeuwen C. (2019) The Impact of Possible Decadal-Scale Cold Waves on Viticulture over Europe in a Context of Global Warming. *Agronomy*, 9, 397; doi:10.3390/agronomy9070397.

Reviewer #2 (Remarks to the Author):

The revised version of the paper has been substantially improved. The AMOC will either overshoot or undershoot depending on when negative emissions are applied. Physically, this means how much

warm/salty water is stored in the subtropics during the weakening phase and the hydrological sensitivity of the model. I now recommend the paper accepted for publication with minor revisions.

1. I suggest we tune down the direct quantitative relevance to policies and reality. The key point is the system's inertial reaction to climate forcing pathways. It's a "possibility" with a lot of uncertainties.

2. I find the caption of Fig.1 confusing. I suggest you add something like: the top row shows results from the B-experiments and the bottom row the overshoot experiments. It would be better to explicitly describe the B-experiments. As this is the key figure for the paper, please make it easier for people to understand.

Good work and an interesting paper!

I don't need to see the revised version again.